# Memristive Circuit Design of Nonassociative Learning under Different Emotional Stimuli

Junwei Sun [1,*], Linhao Zhao [1], Shiping Wen [2] and Yanfeng Wang [1]

1   School of Electrical and Information Engineering, Zhengzhou University of Light Industry, Zhengzhou 450002, China
2   Australia AI Institute, University of Technology Sydney, Ultimo, NSW 2007, Australia
*   Correspondence: junweisun@yeah.net

**Abstract:** Most memristor-based circuits only consider the mechanism of nonassociative learning, and the effect of emotion on nonassociative learning is ignored. In this paper, a memristive circuit that can realize nonassociative learning under different emotional stimuli is designed. The designed circuit consists of stimulus judgment module, habituation module, sensitization module, emotion module. When different stimuli are applied, habituation or sensitisation is formed based on the intensity and nature of the stimuli. In addition, the influence of emotion on nonassociative is considered. Different emotional stimuli will affect the speed of habituation formation and strong negative stimuli will lead to sensitization. The simulation results on PSPICE show that the circuit can simulate the above complex biological mechanism. The memristive circuit of nonassociative learning under different emotional stimuli provides some references for brain-like systems.

**Keywords:** memristor; nonassociative learning; habituation; sensitization; memristive circuit





## 1. Introduction

Recently, brain-like intelligence has received increasing attention and developed rapidly [1–5]. It is inspired by neural mechanisms of the brain and cognitive–behavioral mechanisms and is implemented through software and hardware synergy. For hardware implementation [6–9], integration and power consumption have to be taken into account. Therefore, it is crucial to find a suitable device to build a brain-like intelligence circuit. Memristor is the fourth basic circuit element besides resistor, capacitor and inductor, which represents the relationship between magnetic flux and electric charge. Memristor was predicted by Professor Leon Chua in 1971 [10], and was developed by HP researchers in 2008 [11]. Memristor is a nonlinear device with characteristics similar to the synapses of the biological brain, which can be used as an electronic synapse to effectively simulate the learning and memory functions of the biological brain [12–21]. At present, memristors are widely used in secure communications, pattern recognition, dynamic analysis of chaos system, artificial intelligence computer and other fields [22–26].

Associative learning and nonassociative learning are two important ways of biological learning. Associative learning refers to the learning process realized by the connection between more than two central excitations in the brain caused by two or more stimuli. Associative learning is mainly divided into classical conditioning and operant conditioning and is widely studied. Nonassociative learning refers to that behavior changes are only caused by a single pattern of stimulus repetition. Nonassociative learning is mainly divided into habituation and sensitization [27–29]. Habituation refers to the phenomenon that the spontaneous response is weakened or disappeared under repeated weak stimuli. Sensitization refers to the phenomenon that the spontaneous response gradually increases under repeated strong harmful stimulus. At present, many scholars have realized nonassociative learning through memristor. A synapse-like device based on memristor was proposed in [30], which simulated the Aplysia gill-withdrawal reflex and realized the habituation and

dishabituation behaviors. Nonassociative learning was simulated using a ZnO nanowire memristor in [31], which the memristor shows habituation and sensitization behaviors under electrical and optical stimuli. A nonassociative learning circuit based on memristor was designed in [32] to implement habituation and sensitization behaviors. The generation and influence of emotions have also attracted the attention of many scholars. A memristive circuit of emotional brain-like emotional learning and generation was proposed in [33], which can generate different emotions in 2-D emotional space according to multi-modal information. A memristor-based neural network circuit of memory with emotional homeostasis was designed in [34], which realizes the automatic regulation of emotional neurons. The above works only consider the mechanism of nonassociative learning and the influence of emotion on nonassociative learning is ignored.

Nonassociative learning is affected by the intensity and nature of stimulus [28,35]. The nature of stimulus means that stimulus can be divided into positive stimulus, neutral stimulus and negative stimulus. The intensity of stimulus means that stimulus can be divided into weak stimulus and strong stimulus. When the applied stimulus is a weak stimulus, positive stimulus promotes the formation of habituation and negative stimulus inhibits the formation of habituation. When the applied emotional stimulus is a strong harmful stimulus, sensitization is formed under negative emotional stimulus. In this paper, a memristive neural network circuit that can realize nonassociative learning under different emotional stimuli is proposed. It implements the functions such as habituation, dishabituation, sensitization, and emotion generation. It also realizes the influence of different types of stimulus on habituation and sensitization.

Compared with work [30–34], this work has some advantages in the following aspects. First, the work in this paper implements habituation and sensitization. In addition, habituation under different frequency stimuli is also realized. Habituation and sensitization are achieved through the habituation module and sensitization module, which can achieve more functions compared to the previous works [30,31]. Second, the work in this paper implements nonassociative learning under different emotional stimuli by combining nonassociative learning and emotion. Nonassociative learning under different emotional stimuli is achieved through the habituation module, emotion module, stimulus judgment module and sensitization module, which is more bionic than the previously proposed neural network circuits [32–34].

The rest of the paper is arranged as follows. Section 2 introduces a threshold memristor model. Section 3 introduces the biological mechanism of nonassociative learning under different emotional stimuli. Section 4 introduces the circuit structure and the functions of each module. In Section 5, the simulation results of the circuit are analyzed in detail. Section 6 draws some conclusions.

## 2. Memristor Model with Threshold

Various memristor models have been designed in [36–38]. The basic model of memristor is shown in Figure 1. In this paper, a voltage-controlled threshold memristor model based on the experimental data of AIST-based memristor is used. The model is as follows:

$$M(t) = R_{ON}\frac{w(t)}{D} + R_{OFF}(1 - \frac{w(t)}{D}) \tag{1}$$

where $M(t)$ is the memristance, $R_{on}$ is the resistance at highly doped, $R_{off}$ is the resistance at low doped, $w(t)$ is the width of the highly doped region, $D$ is the full thickness of memristive material. The derivative of the state variable $w(t)$ is

$$\frac{dw(t)}{dt} = \begin{cases} \mu_v \frac{R_{ON}}{D} \frac{i_{off}}{i(t)-i_0} f(w(t)), & v(t) > V_{T+} > 0 \\ 0, & V_{T-} \leqslant v(t) \leqslant V_{T+} \\ \mu_v \frac{R_{ON}}{D} \frac{i(t)}{i_{on}} f(w(t)), & v(t) < V_{T-} < 0 \end{cases} \tag{2}$$

where $\mu_v$ stands for the ionic mobility, $i_0$, $i_{on}$ and $i_{off}$, are constants. $V_{T+}$ and $V_{T-}$ are positive and negative threshold voltages, respectively. $i(t)$ and $v(t)$ are the current and voltage of the memristor, respectively. $f(w(t))$ is given as a window function.

$$f(w(t)) = 1 - (\frac{2w(t)}{D} - 1)^{2p} \tag{3}$$

where $p$ is a positive integer.

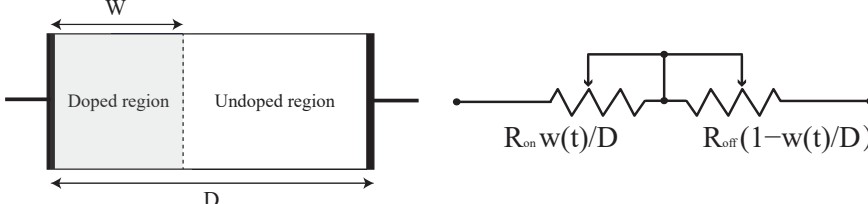

**Figure 1.** Basic model of the memristor.

If a positive voltage beyond its positive threshold is applied to the memristor, the value of the memristor will decrease. As shown in Figure 2, the greater the positive voltage is, the slower the memristance decreases. The opposite case will happen when the voltage is a negative voltage. The various parameters of the four memristors are shown in Table 1. All the simulation processes in this paper are implemented by PSPICE.

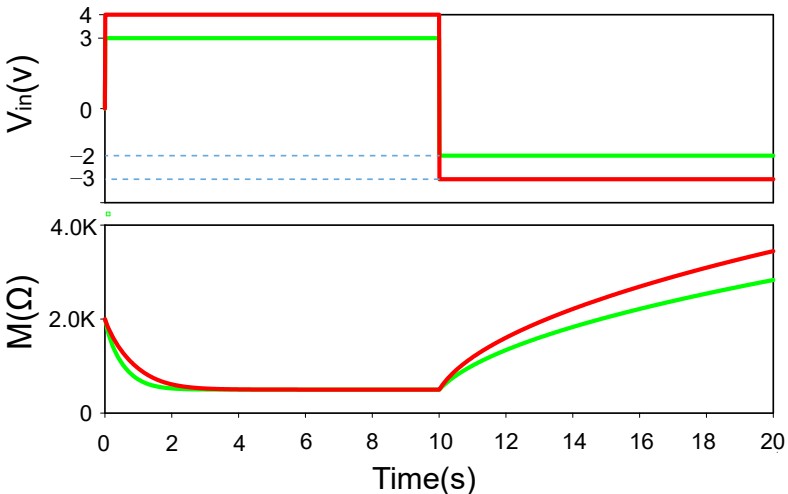

**Figure 2.** The change in memristance when the memristors are applied by different voltages. $V_{in}$ represents input voltage. $M$ represents the memristance of memristor.

**Table 1.** Parameters setting of memristors.

| Parameters | $M_1$ | $M_2$ | $M_3$ | $M_4$ |
|---|---|---|---|---|
| $D$ (nm) | 3 | 3 | 3 | 3 |
| $R_{ON}(\Omega)$ | 500 | 100 | 500 | 100 |
| $R_{OFF}(\Omega)$ | 6k | 1k | 1k | 1k |
| $R_{init}(\Omega)$ | 4k | 1k | 500 | 1k |
| $V_{T+}(v)$ | 3.5 | 1 | 1.1 | 1 |
| $V_{T-}(v)$ | −0.5 | −0.1 | −0.5 | −0.099 |
| $\mu_v(\mathrm{m^2 s^{-1} \Omega^{-1}})$ | $6 \times 10^{-19}$ | $3 \times 10^{-17}$ | $5 \times 10^{-18}$ | $1 \times 10^{-17}$ |
| $i_{on}$ (A) | 1 | 1 | 1 | 1 |
| $i_{off}$ (A) | $1 \times 10^{-5}$ | $1 \times 10^{-5}$ | $1 \times 10^{-5}$ | $1 \times 10^{-5}$ |
| $i_0$ (A) | $1 \times 10^{-3}$ | $1 \times 10^{-3}$ | $1 \times 10^{-3}$ | $1 \times 10^{-3}$ |
| $p$ | 10 | 10 | 10 | 10 |

### 3. Nonassociative Learning under Different Emotional Stimuli

Nonassociative learning under different emotional stimuli in this paper is illustrated in Figure 3. Nonassociative learning is mainly divided into habituation and sensitization. Habituation refers to the phenomenon that the spontaneous response is weakened with stimulus repetition. Sensitization refers to the phenomenon that the spontaneous response gradually increases with strong harmful stimulus repetition. Habituation is mainly divided into two categories, including behavioral habituation and emotional habituation. The stimulus that causes behavioral habituation is not emotional, but the stimulus that causes emotional habituation is emotional. $PE$, $NE_1$, and $NE_2$ are emotional stimuli, where the stimulus intensity of $NE_2$ is higher than that of $NE_1$. Positive feeling is generated when positive stimulus $PE$ is applied, and negative feeling is generated when negative stimulus $NE_1$ or $NE_2$ is applied.

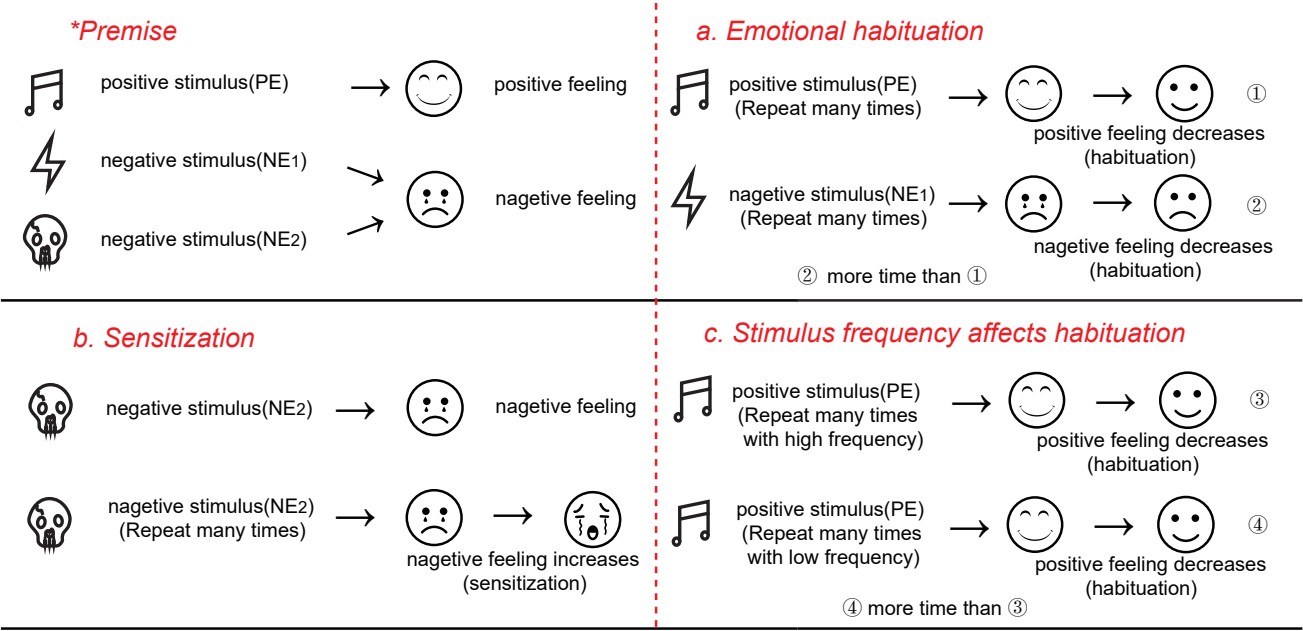

**Figure 3.** Illustration of nonassociative learning under different emotional stimuli.

Emotional habituation is described in process *a*. When different emotional stimuli are applied, the brain will produce different feelings. When positive stimulus is applied at the beginning, strong positive feeling is generated. The degree of the positive feeling will slowly decline with positive stimulus repetition, which indicates that emotional habituation occurs. A similar phenomenon occurs with negative stimulus repetition. The difference is that it takes longer to form habituation under a negative stimulus than under a positive stimulus. The phenomenon shows that emotional habituation shows a negative bias, which maintains a high degree of arousal of negative emotional stimulus and avoids the harm caused by the negative stimulus.

Sensitization is described in process *b*. When negative stimulus $NE_2$ is applied at the beginning, strong negative feeling is generated. Because $NE_2$ is a strong negative stimulus, habituation will not occur with the negative stimulus $NE_2$ repetition. On the contrary, sensitization will occur with the negative stimulus $NE_2$ repetition. The degree of negative feeling will gradually increase when the negative stimulus $NE_2$ is applied, which indicates that sensitization occurs. The effect of stimulus frequency on habituation is described in process *c*. When high frequency stimulus is applied, the time for habituation formation is shorter than when a low frequency stimulus is applied. The phenomenon shows that high frequency stimulus can cause more intense habituation than low frequency stimulus.

## 4. Circuit Structure and Module Design

The diagram of circuit design is shown as in Figure 4. When the input signal $N_1$ and emotional signal $N_2$ are applied, the stimulus judgment module will judge the nature of $N_1$ according to the voltage of $N_1$ and $N_2$. The formation of habituation is promoted when $N_1$ is positive repeated stimulus. In addition, the dotted line indicates that habituation is also affected by the frequency of input stimulus. The formation of habituation is inhibited when $N_1$ is weak negative repeated stimulus. When $N_1$ is strong negative repeated stimulus, sensitization will occur with negative stimulus $N_1$ repetition.

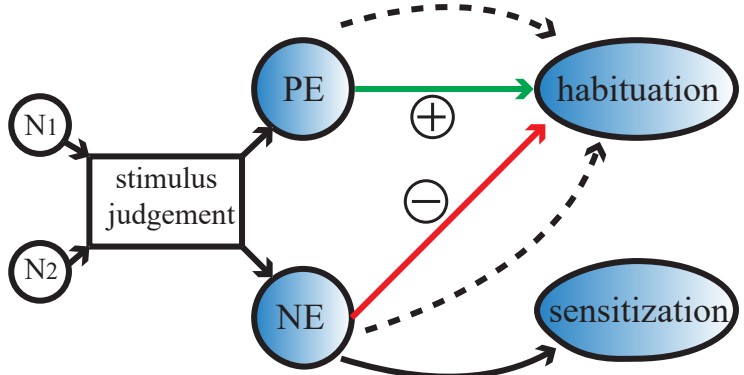

**Figure 4.** Diagram of circuit design. $N_1$ is the input signal. $N_2$ is the emotional signal. $PE$ represents that the input signal $N_1$ is positive stimulus. $NE$ represents that the input signal $N_1$ is negative stimulus.

### 4.1. Stimulus Judgment Module

The stimulus judgment module is shown in Figure 5. The function of the stimulus judgment module is to transmit the input signal $N_1$ to the habituation module or the sensitization module according to the voltage of the input signal $N_1$ and emotional signal $N_2$. $OP_1$ and $OP_2$ are two comparators. $OP_1$ is used to judge the intensity of input stimulus $N_1$. When the voltage of $N_1$ is lower than $V_1$, $OP_1$ outputs low level, which indicates that $N_1$ is weak stimulus. When the voltage of $N_1$ is higher than $V_1$, $OP_1$ outputs high level, which indicates that $N_1$ is strong stimulus. $OP_2$ is used to judge the nature of input stimulus $N_1$. When the voltage of $N_2$ is lower than $V_2$, $OP_2$ outputs low level, which indicates that $N_1$ is nagetive stimulus. When the voltage of $N_2$ is higher than $V_2$, $OP_2$ outputs high level, which indicates that $N_1$ is positive or neutral stimulus. Logical unit is composed of $D_1$, $D_2$, $D_3$, $D_4$, $S_1$ and $S_2$, which can determine the voltage of $a$ and $b$ according to the output of $OP_1$ and $OP_2$. The voltage of $a$ and $b$ is shown in Table 2. When $V_b = N_1$, the input signal $N_1$ is negative strong stimulus, otherwise $N_1$ is other stimulus.

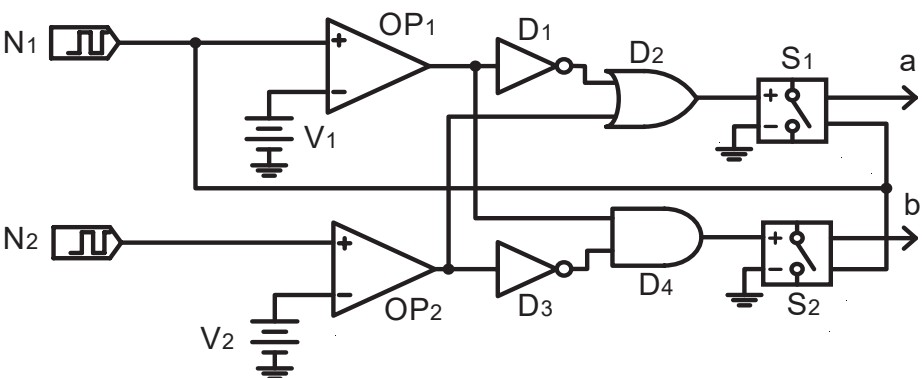

**Figure 5.** Stimulus judgement module. $OP_1$ and $OP_2$ are comparators. $V_1 = 2.99$ V, $V_2 = -0.1$ V.

**Table 2.** The voltage of *a*, *b*.

| $OP_1$ | $OP_2$ | $V_a$ | $V_b$ |
|---|---|---|---|
| low level | low level | $N_1$ | 0 |
| low level | high level | $N_1$ | 0 |
| high level | low level | 0 | $N_1$ |
| high level | high level | $N_1$ | 0 |

### 4.2. Habituation Module

The habituation module is shown in Figure 6. When $V_a$ is initially input, the memristance of $M_3$ remains unchanged because the amplitude of $V_a$ fails to reach the positive threshold voltage of $M_3$. The output of $OP_8$ is $V_{OP_8} = -(R_{12}/M_3) \times V_{SUM_2}$. $ABS_2$ is an absolute value module, which can obtain the absolute value of $V_{OP_8}$. When $V_a$ is high level, voltage-controlled switch $S_{10}$ is closed and the output of $ABS_2$ is transferred to $N_3$, where $N_3$ represents the output signal of the habituation module. The output voltage of $N_3$ exceeds the closed voltage of $S_{12}$ because the memristance of $M_3$ is unchanged. The function of $OP_9$ is to sum the voltage, where the output of $OP_9$ is $V_{OP_9} = V_x + V_y + V_z$. $SUM_4$ is sum component. $V_{12}$ is applied to $M_3$ through $OP_9$ and $SUM_4$. $V_{12}$ exceeds the positive threshold voltage of $M_4$ and the memristance of $M_4$ is gradually reduced. The output of $ABM_2$ is $V_{ABM_2} = -V_{IN_2}/V_{IN_1} = M_4/1000$. $V_{ABM_2}$ is gradually reduced due to the decrease of the memristance of $M_4$. $OP_{11}$ is voltage comparator and $OP_{11}$ outputs high level when the input voltage is lower than $V_{14}$. When $V_{ABM_2}$ is less than $V_{14}$, $OP_{11}$ outputs high level. Voltage-controlled switch $S_{10}$ is closed and $V_{14}$ is transmitted to $S_{15}$ as feedback voltage $F_1$. When $V_a$ is low level, voltage-controlled switch $S_{15}$ is closed and feedback signal $F_1$ is applied to $M_3$ through $SUM_2$. $F_1$ is less than the negative threshold voltage of $M_3$ and the memristance of $M_3$ is gradually increased. When $V_a$ is high level, the output of $OP_8$ and $N_3$ are gradually reduced, which indicates that habituation occurs. When $N_3$ is low level, voltage-controlled switch $S_{11}$ is closed. $V_{11}$ is applied to $M_3$ through $OP_9$ and $SUM_4$. $V_{12}$ is less than the negative threshold voltage of $M_4$ and the memristance of $M_4$ is gradually increased. When the output of $ABM_2$ exceeds the threshold of comparator $OP_{11}$, $F_1$ will disappear. When the amplitude of $V_a$ is increased, $V_a$ exceeds the positive threshold voltage of $M_3$ and the memristance of $M_3$ is decreased. When $V_a$ is high level, the output of $ABS_2$ and $N_3$ are gradually increased, which indicates that dishabituation occurs.

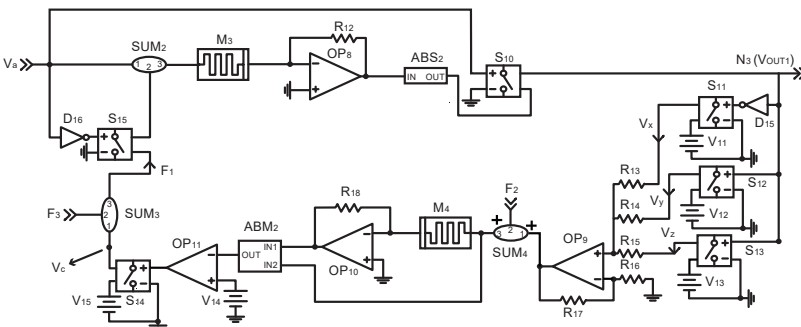

**Figure 6.** Habituation module. $F_1$, $F_2$ and $F_3$ are feedback signals. $S_{11}$, $S_{12}$, $S_{13}$, $S_{14}$, $S_{15}$ and $S_{10}$ are the voltage-controlled switches. $V_{11} = -0.1$ V, $V_{12} = 4$ V, $V_{13} = -6.5$ V, $V_{14} = 0.2$ V, $V_{15} = -1$ V. The closed voltage of $S_{14}$ and $S_{15}$ is 2 V. The closed voltage of $S_{11}$ and $S_{12}$ is 1 V. The closed voltage of $S_{10}$ is 0.9 V. The closed voltage of $S_{13}$ is 2.1 V. $R_{12} = 1$ kΩ, $R_{18} = 100$ Ω, $R_{13} = R_{14} = R_{15} = R_{17} = 20$ kΩ, $R_{16} = 10$ kΩ.

### 4.3. Emotion Module

The emotion module is shown in Figure 7. The function of emotion module is to generate different feedback signals according to the emotion signal ($N_2$). Feedback signals affect the time of forming habituation. Habituation is more easily formed with positive

stimulus repetition, while habituation is less easily formed with negative stimulus repetition. $N_2 = 1$ V indicates positive emotion, $N_2 = -1$ V indicates negative emotion and $N_2 = 0$ V indicates neutral emotion. When positive stimulus is applied, $OP_7$ outputs a feedback signal ($F_2$). The change in the memristance of $M_4$ is influenced by $F_2$. When $V_a$ is high level, the memristance of $M_4$ decreases faster. When $V_a$ is low level, the memristance of $M_4$ increases slower. The memristance of $M_4$ decreases faster than in the absence of $F_2$ and the time to generate $F_1$ is reduced. $OP_{12}$ outputs a feedback signal $F_3$ due to the generation of $F_1$. The voltage of $F_1$ is increased due to the presence of $F_3$, which makes the memristance of $M_3$ rise faster. The time to form habituation is also reduced. When negative stimulus is applied, $OP_7$ also outputs a feedback signal ($F_2$). The memristance of $M_4$ decreases slower than in the absence of $F_2$ and the time to generate $F_1$ is increased. $OP_{12}$ outputs a feedback signal $F_3$ to reduce the voltage of $F_1$, which makes the memristance of $M_3$ rise slower. The time to form habituation is increased. Feedback signals cannot be generated when neutral stimulus is applied, indicating that neutral stimulus has no effect on habituation.

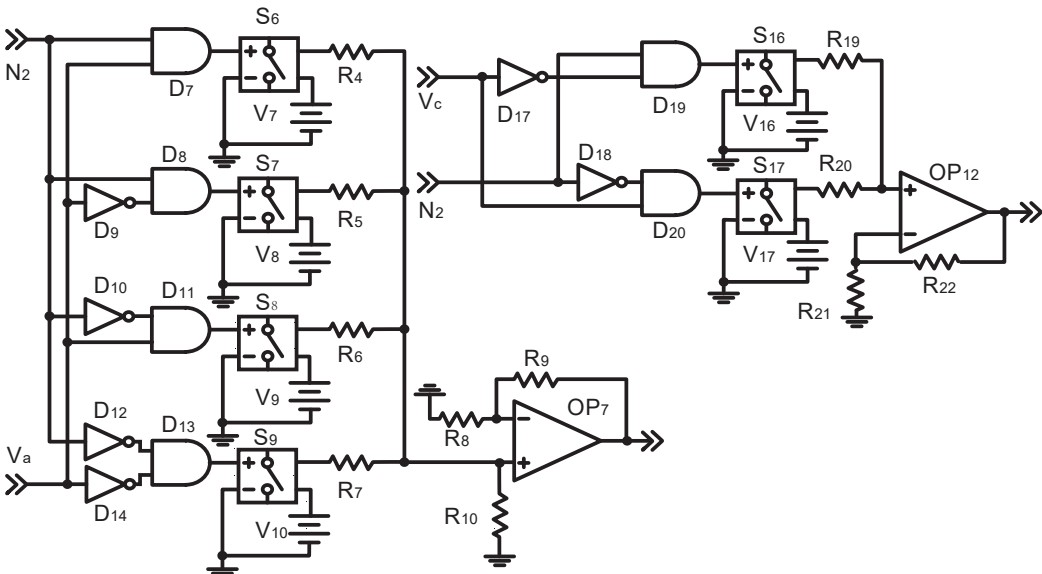

**Figure 7.** Emotion module. $V_7 = -0.5$ V, $V_8 = 0.01$ V, $V_9 = 0.5$ V, $V_{10} = -0.01$ V, $V_{16} = -0.2$ V, $V_{17} = 0.2$ V. $R_4 = R_5 = R_6 = R_7 = R_9 = R_{10} = R_{19} = R_{20} = R_{21} = R_{22} = 20$ kΩ, $R_8 = 5$ kΩ. $D_7$ is a AND gate to control $S_6$. $D_8$ and $D_9$ control $S_7$. $D_{10}$ and $D_{11}$ control $S_8$. $D_{12}$, $D_{13}$ and $D_{14}$ control $S_9$. $D_{17}$ and $D_{19}$ control $S_{16}$. $D_{18}$ and $D_{20}$ control $S_{17}$. $OP_{12}$, $R_{19}$, $R_{20}$, $R_{21}$, $R_{22}$ form a voltage summation unit. $OP_7$, $R_4$, $R_5$, $R_6$, $R_7$, $R_8$, $R_9$ and $R_{10}$ form a voltage summation unit.

### 4.4. Sensitization Module

The sensitization module is shown in Figure 8. When $V_b$ is initially input, the memristance of $M_1$ remains unchanged because the amplitude of $V_b$ fails to reach the positive threshold voltage of $M_1$. $M_1$, $R_1$ and OPE module are used to realize proportional operation. The output of $V_d$ is $V_d = (1 - R_1/M_1) \times V_{SUM_1}$. $ABS_1$ is an absolute value module, which can obtain the absolute value of $V_d$. When $V_d$ is high level, voltage-controlled switch $S_3$ is closed and the output of $ABS_1$ is transmitted to $N_4$. Comparator is composed of $OP_3$ and $OP_4$. The output of $ABM_1$ is $V_{ABM_1} = -V_{IN_2}/V_{IN_1} = M_2/1000$. When $V_4 < V_{OUT_2} < V_3$, $D_6$ outputs high level. The output of $D_6$ exceeds the positive threshold voltage of $M_2$ and the memristance of $M_2$ is gradually reduced. The output of $ABM_1$ is also gradually reduced. When the output of $ABM_1$ is less than $V_5$, the comparator $OP_6$ outputs high level. Voltage-controlled switch $S_3$ is closed and the feedback voltage $F_4$ is generated, where $F_4 = V_6$. When $V_b$ is low level, feedback voltage $F_4$ is applied to $M_1$ through $SUM_1$. $F_4$ is less than the negative threshold voltage of $M_1$ and the memristance of $M_1$ is gradually increased. When $V_b$ is high level, the output of $N_4$ is gradually increased, which indicates that sensitization occurs. When the voltage of $N_4$ exceeds $V_3$, $D_6$ outputs low level and

feedback signal $F_4$ cannot be generated. The output of $N_6$ remains unchanged, which means that the sensitization is completed.

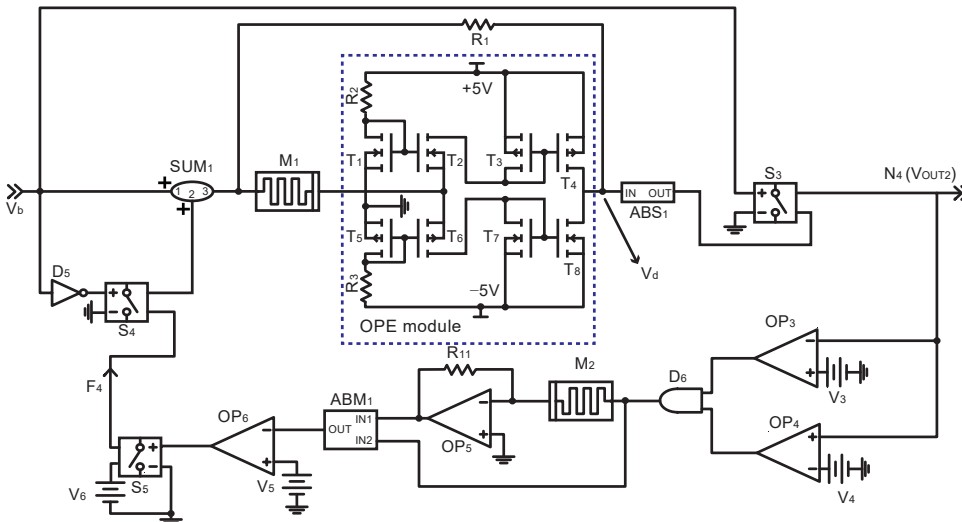

**Figure 8.** Sensitization module. $F_4$ is feedback signal. $S_3$, $S_4$, and $S_5$ are the voltage-controlled switches. The closed voltage of $S_5$ is 2 V. The closed voltage of $S_3$ is 0.9 V. The closed voltage of $S_4$ is 2 V. $R_1 = 1\,\text{k}\Omega$, $R_{11} = 100\,\Omega$, $V_3 = 3.5\,\text{V}$, $V_4 = 2\,\text{V}$.

### 4.5. Complete Circuit

The complete circuit is shown in Figure 9, in which stimulus judgment module, habituation module, sensitization module, emotion module are connected with each other. The designed circuit realizes the functions of habituation, dishabituation, sensitization, emotion generation, frequency-dependent habituation and so on, which are verified by PSPICE.

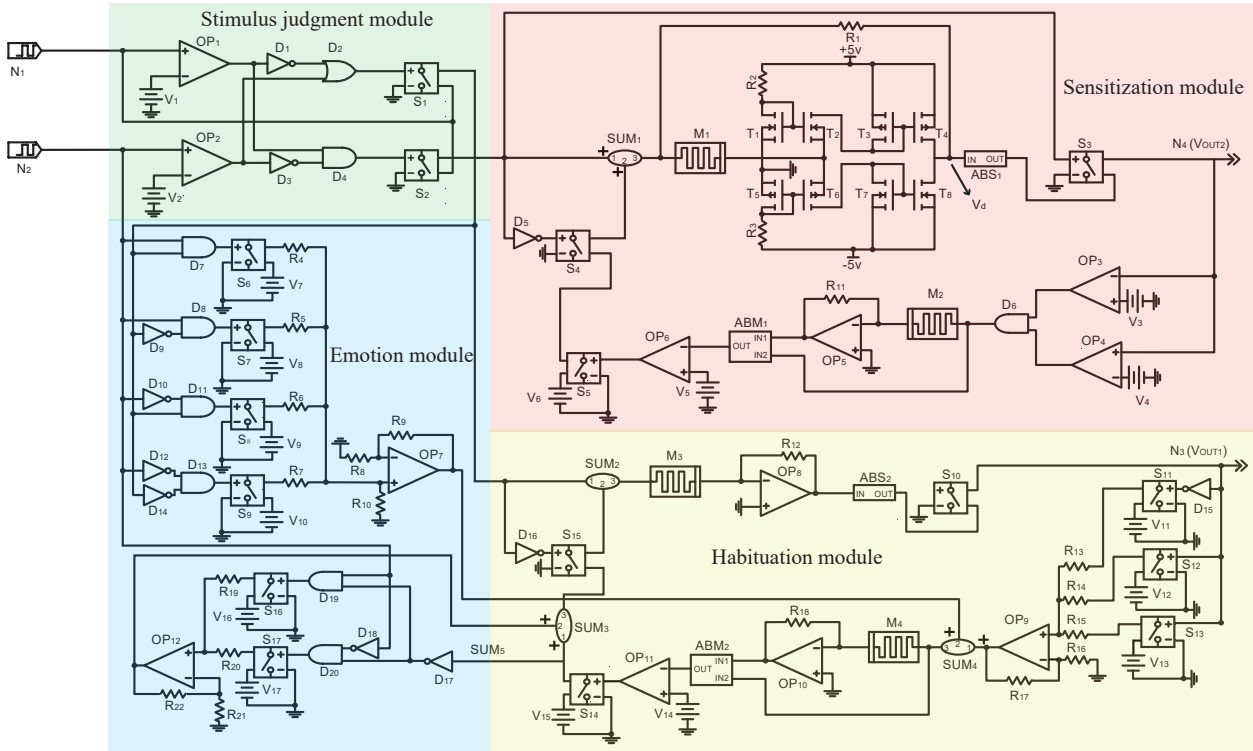

**Figure 9.** Complete circuit.

## 5. Implementation and Simulation of Circuit

### 5.1. Habituation and Dishabituation

#### 5.1.1. Habituation

The simulation results of habituation are shown in Figure 10. The voltage of emotion signal $N_1$ is 0 V, which indicates that the input signal $N_1$ is neutral stimulus. From 0 s to 14 s, the feedback signal $F_1$ is not generated. When $N_1$ is high level, the output of $N_3$ is $V_{N_3} = V_{ABS_1} = (R_{12}/M_3) \times V_{N1} = 2$ V, where the memristance of $M_3$ is 0.5 kΩ. At 14 s, the memristance of $M_4$ is reduced to 0.2 kΩ and the feedback signal $F_1$ is generated. The voltage of $F_1$ is less than the negative threshold voltage of $M_3$. When $F_1$ is applied to $M_3$, the memristance of $M_3$ is gradually increased. The voltage of $N_3$ is also gradually decreased with the increase of the memristance of $M_3$. At 34 s, the memristance of $M_3$ increases to 1 kΩ. The voltage of $N_3$ is reduced to 1 V and remains unchanged, which shows that habituation is formed. The time of habituation under the neutral stimulus is 34 s.

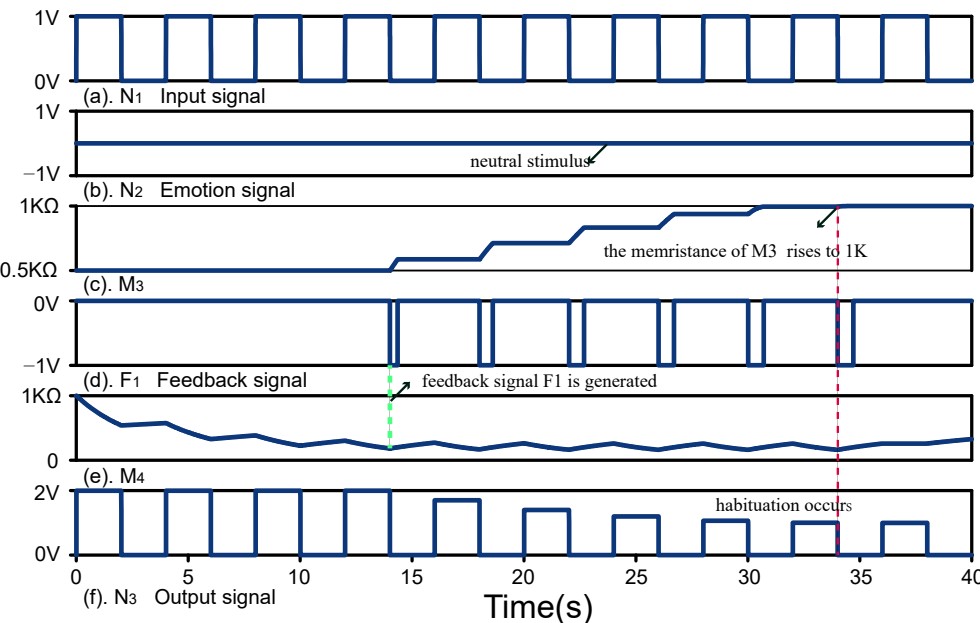

**Figure 10.** Habituation . (**a**) Input signal. (**b**) Emotion signal. (**c**) Memristance of $M_3$. (**d**) Feedback signal $F_1$. (**e**) Memristance of $M_4$. (**f**) Output signal $N_3$ .

#### 5.1.2. Dishabituation

The simulation results of dishabituation are shown in Figure 11. At 48 s, a strong stimulus $N_1$ is applied, where the voltage of $N_1$ is 3 V. Sensitization cannot occur because $N_1$ is a neutral stimulus. The memristance of $M_3$ decreases rapidly from 1 kΩ to 0.5 kΩ and the memristance of $M_4$ rises rapidly to 1 kΩ. At 52 $s$, the voltage of $N_1$ is reduced from 3 V to 1 V and the output of $N_3$ is reduced from 6 V to 2 V, which indicates that dishabituation is completed.

### 5.2. Emotional Habituation

Habituation is influenced by emotional stimuli. When different repeated emotional stimuli are applied, the time taken to form habituation is also different. Positive stimulus promotes the formation of habituation, while negative stimulus inhibits the formation of habituation.

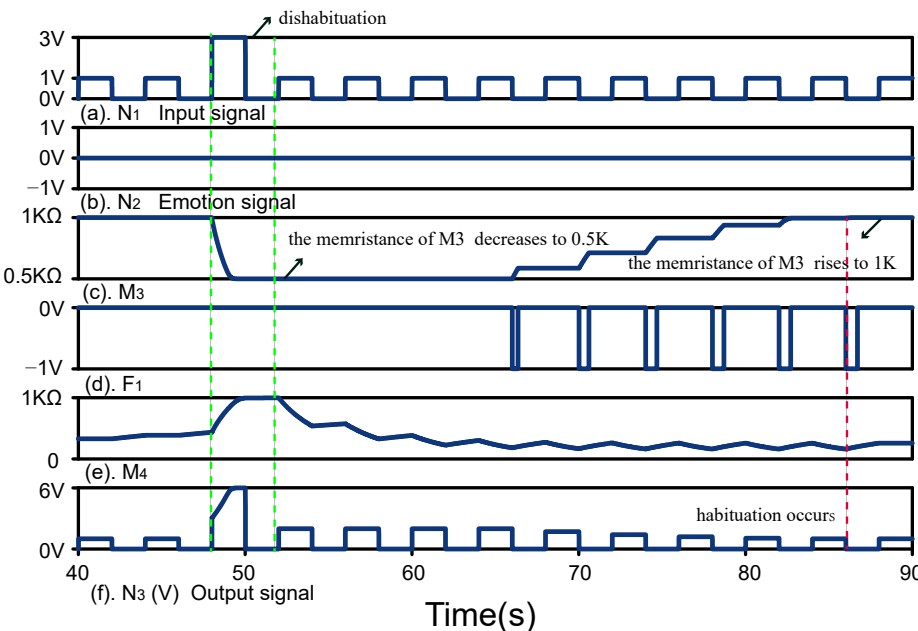

**Figure 11.** Dishabituation . (**a**) Input signal. (**b**) Emotion signal. (**c**) Memristance of $M_3$. (**d**) Feedback signal $F_1$. (**e**) Memristance of $M_4$. (**f**) Output signal $N_3$ .

5.2.1. Habituation Under Positive Stimulus

The simulation results of habituation under positive stimulus are shown in Figure 12. The voltage of $N_2$ is 1 V, which indicates that $N_1$ is positive stimulus. When positive stimulus is applied, feedback signals $F_2$ and $F_2$ are generated. When $N_1$ is high level, the voltage of $F_2$ is $-0.5$ V. The voltage applied to $M_4$ is reduced from 4 V to 3.5 V due to the effect of $F_2$, which causes the memristance of $M_4$ to drop faster. At 9 s, the memristance of $M_4$ drops to 0.2 kΩ. When $N_1$ is low level, the voltage of $F_2$ is 0.01 V. The voltage applied to $M_4$ is increased from $-0.1$ V to $-0.09$ V due to the effect of $F_2$, which is greater than the negative threshold voltage of $M_4$. The memristance of $M_4$ remains unchanged. The time taken for the feedback voltage $F_1$ to be generated is reduced, which also reduces the time for habituation formation. When the feedback signal $F_1$ appears, the feedback signal $F_3$ is generated. Feedback signal $F_1$ decreases from $-1$ V to $-1.2$ V due to the effect of $F_3$, which causes the memristance of $M_3$ to rise more quickly. At 15 s, the memristance of $M_3$ increases from 0.5 kΩ to 1 kΩ and the voltage of $N_3$ decreases from 2 V to 1 V which indicates that habituation is formed. The time of habituation under the positive stimulus is 15 s.

5.2.2. Habituation Under Negative Stimulus

The simulation results of habituation under negative stimulus are shown in Figure 13. The voltage of $N_2$ is $-1$ V, which indicates that $N_1$ is negative stimulus. When $N_1$ is high level, the voltage of $F_2$ is 0.5 V. The voltage applied to $M_4$ is increased from 4 V to 4.5 V due to the effect of $F_2$, which causes the memristance of $M_4$ to drop slower. When $N_1$ is low level, the voltage of $F_2$ is $-0.01$ V, which causes the memristance of $M_4$ to drop faster. At 17.5 s, the memristance of $M_4$ drops to 0.2 kΩ. The voltage of $F_3$ under negative stimulus is 0.2 V. Feedback signal $F_1$ increases from $-1$ V to $-0.8$ V due to the effect of $F_3$, which causes the memristance of $M_3$ to rise more slowly. It also takes more time for the voltage of $N_3$ to decrease from 2 V to 1 V. At 60 s, the voltage of $N_3$ is reduced from 2 V to 1 V. The time of habituation under the negative stimulus is 60 s.

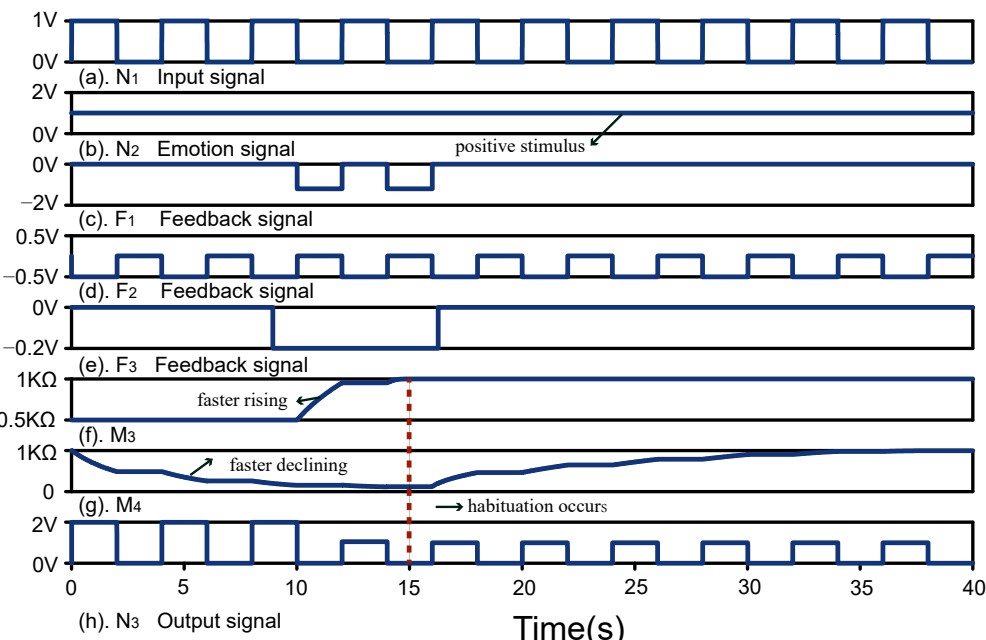

**Figure 12.** Habituation under positive stimulus. (**a**) Input signal. (**b**) Emotion signal. (**c**) Feedback signal $F_1$. (**d**) Feedback signal $F_2$. (**e**) Feedback signal $F_3$. (**f**) Memristance of $M_3$. (**g**) Memristance of $M_4$. (**h**) Output signal $N_3$.

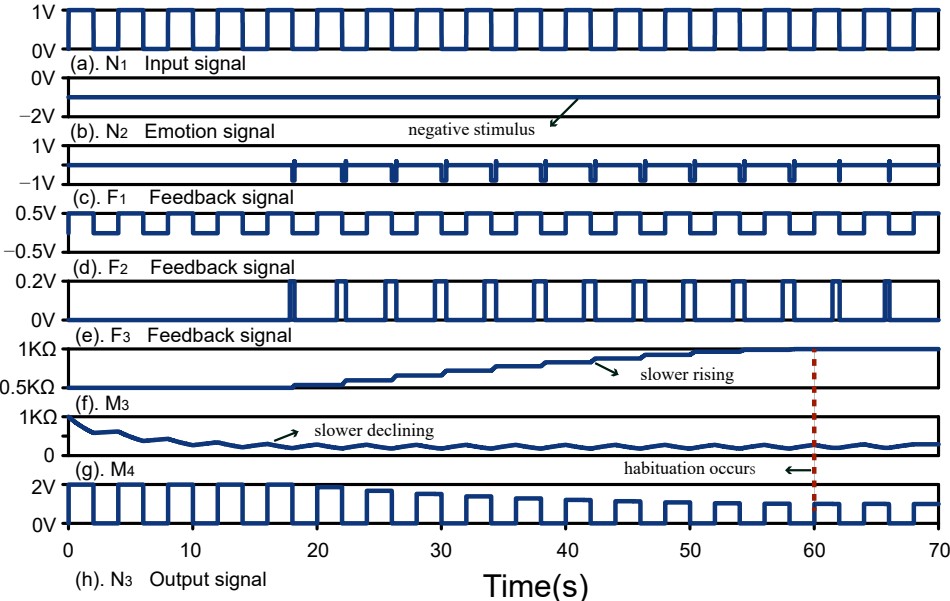

**Figure 13.** Habituation under negative stimulus. (**a**) Input signal. (**b**) Emotion signal. (**c**) Feedback signal $F_1$. (**d**) Feedback signal $F_2$. (**e**) Feedback signal $F_3$. (**f**) Memristance of $M_3$. (**g**) Memristance of $M_4$. (**h**) Output signal $N_3$.

### 5.3. Frequency-Dependent Habituation

In addition to being influenced by emotion, habituation is also affected by the frequency of stimulus. As shown in Figure 14, frequency of $N_1$ is increased. When $N_1$ is low level, the memristance of $M_4$ is increased under the effect of $V_{11}$, which is called the callback of the memristance of $M_4$. The callback of the memristance of $M_4$ is reduced due to an increase in the frequency of the input stimulus. The time taken to increase the memristance of $M_3$ from 0.5 kΩ to 1 kΩ is reduced. At 20 s, the memristance of $M_3$ rises to 1 kΩ, which

indicates that habituation is formed. By comparing Figure 10, habituation is more easily formed under high frequency of stimulus.

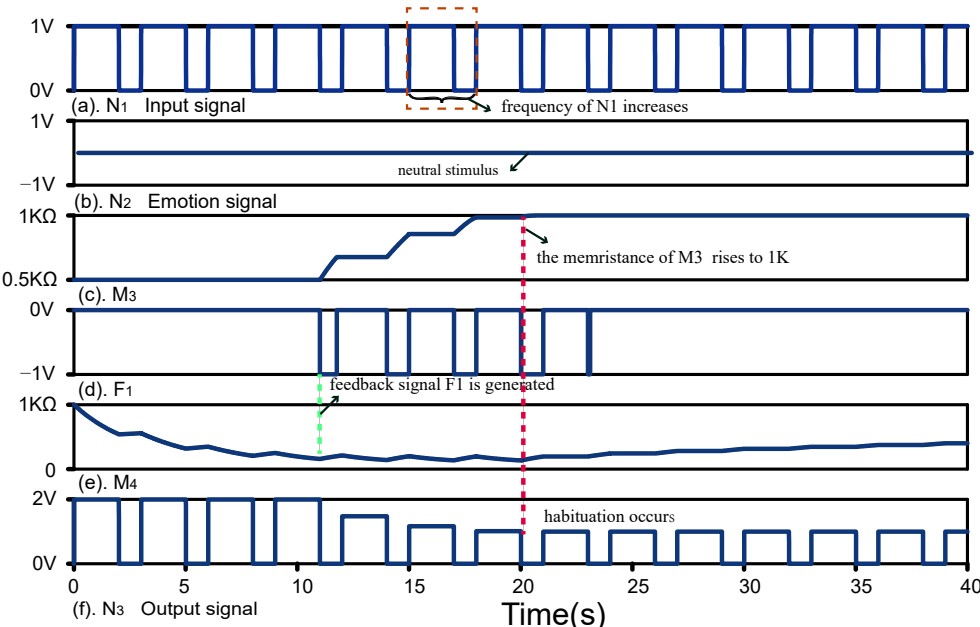

**Figure 14.** Frequency -dependent habituation. (**a**) Input signal. (**b**) Emotion signal. (**c**) Memristance of $M_3$. (**d**) Feedback signal $F_1$. (**e**) Memristance of $M_4$. (**f**) Output signal $N_3$ .

*5.4. Sensitisation*

The simulation results of sensitisation are shown in Figure 15. The voltage of $N_1$ is 3 V and the voltage of $N_2$ is −1 V, which means that $N_1$ is negative strong stimulus. Sensitization is formed with negative stimulus $N_1$ repetition. The voltage of $N_1$ is less than the positive threshold voltage of $M_1$ and the value of $M_1$ remains unchanged. The output of OPE module is $V_d = (1 - R_1/M_1) \times V_{SUM_1} = 2.25$ V, where $R_1 = 1$ kΩ and $M_1 = 4$ kΩ. When $V_4 < V_d < V_3$, $D_6$ outputs high level voltage $V_{high}$ , where $V_{high} = 3.5$ V. The output of $ABM_2$ is $M_2/1000$. $V_{high}$ exceeds the positive threshold value of $M_2$. The value of $M_2$ and the output of $ABM_2$ are reduced. When the output of $ABM_2$ is less than $V_{14}$, $S_{14}$ is closed. When $N_1$ is low level, the feedback voltage $F_4$ is applied to $M_1$ to increase the value of $M_1$. The change in the memristance of $M_1$ leads to the increase of the voltage of $V_d$. The voltage of $N_4$ is also increased, which indicates sensitization occurs.

*5.5. Result Analysis and Comparison Analysis*

As shown in Table 3, a comparison is made with the previous nonassociative learning circuits. The more complex functions of nonassociative learning under different emotional stimuli are realized in this article. Compared with work [31–35], our work realizes the functions of habituation, dishabituation, sensitization, frequency-dependent habituation, emotion generation, emotional habituation, emotional sensitization.

Circuit performance is influenced by process variations and noise sources. In this paper, the voltage variation of $N_3$ is tested by PSpice Monte Carlo simulation. The reliability of the system is affected by the tolerance of devices. The tolerance of $R_{on}$ and $R_{off}$ for all memristor are set to 5%, and the simulation is repeated for 100 times continuously. As shown in Figure 16, the results shows that the circuit system has high reliability.

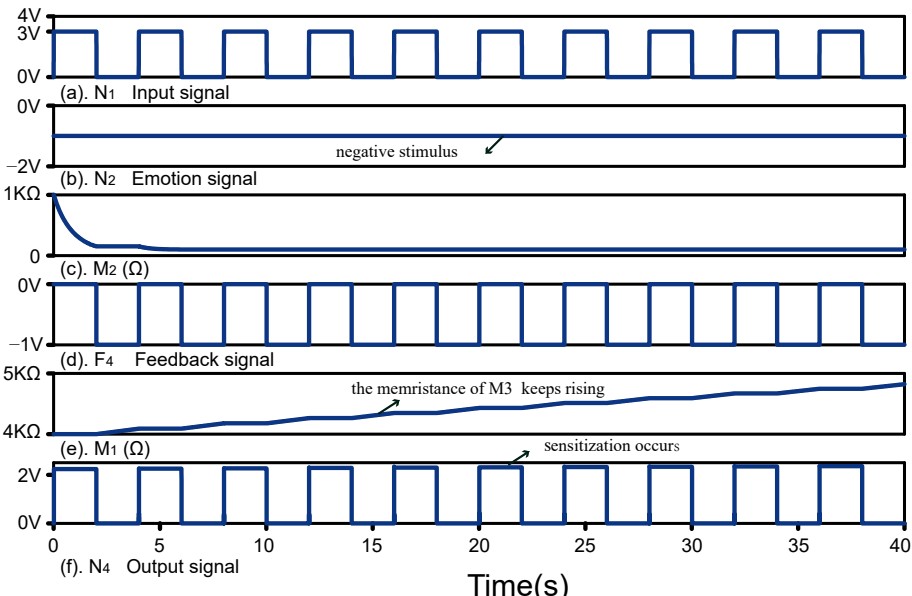

**Figure 15.** Sensitisation. (**a**) Input signal. (**b**) Emotion signal. (**c**) Memristance of $M_2$. (**d**) Feedback signal $F_5$. (**e**) Memristance of $M_1$. (**f**) Output signal $N_4$.

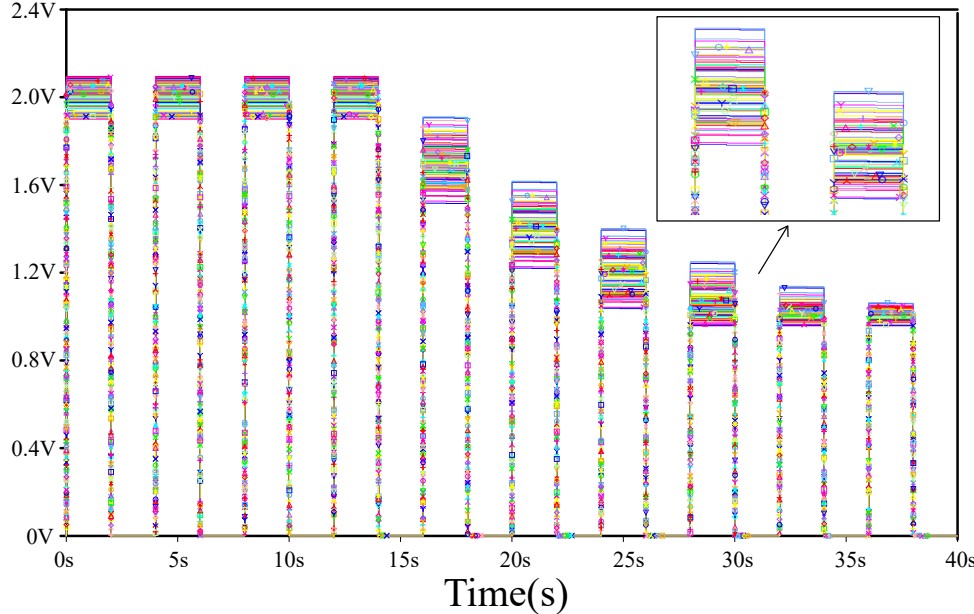

**Figure 16.** Monte Carlo simulation.

**Table 3.** Comparison of functions implemented by several different works.

| Work | Habituation | Dishabituation | Sensitization | Frequency-Dependent Habituation | Emotion Generation | Emotional Habituation | Emotional Sensitization |
|------|-------------|----------------|---------------|----------------------------------|--------------------|-----------------------|-------------------------|
| [30] | √ | × | × | √ | × | × | × |
| [31] | √ | √ | √ | × | × | × | × |
| [32] | × | × | × | × | √ | × | × |
| [33] | × | × | × | × | √ | × | × |
| [34] | √ | √ | √ | √ | √ | √ | √ |
| this work | √ | √ | √ | √ | √ | √ | √ |

## 6. Conclusions

In this paper, a memristive circuit of nonassociative learning under different emotional stimuli has been designed, in which the functions of habituation, dishabituation, frequency-dependent habituation, emotional habituation, and sensitisation have been implemented. The effect of different emotional stimuli on nonassociative learning has been considered, where positive stimulus promotes habituation formation while negative stimuli inhibit habituation formation. In addition, sensitization will occur with strong negative stimulus repetition. Frequency-dependent habituation is also implemented, which enables the designed circuit to be more bionic. However, the circuit designed only considered the effects of emotion and the frequency of stimulus on nonassociative learning, which ignores the effects of other factors on nonassociative learning. For example, the variety of stimuli and different personalities can have an impact on nonassociative learning. In addition, we should concentrate on improving the integration of circuits and implementing more complex biological mechanisms.

**Author Contributions:** Methodology, J.S.; software, L.Z.; validation, Y.W.; formal analysis, L.Z. and Y.W.; investigation, S.W.; resources, J.S.; data curation, Y.W.; writing—original draft, L.Z.; writing—review and editing, J.S.; visualization, S.W.; supervision, J.S.; project administration, S.W.; funding acquisition, Y.W. All authors have read and agreed to the published version of the manuscript.

**Funding:** This work was supported in part by the National Natural Science Foundation of China under Grant 62276239 and 62272424, in part by the Joint Funds of the National Natural Science Foundation of China under Grant U1804262, in part by Henan Province University Science and Technology Innovation Talent Support Plan under Grant 20HASTIT027, in part by Zhongyuan Thousand Talents Program under Grant 204200510003, in part by Zhongyuan Talents Program under Grant ZYYCYU202012154, and in part by Henan Natural Science Foundation–Outstanding Youth Foundation under Grant 222300420095, and in part by the Henan Province Key Research and Development and promotion special project (Science and technology) under Grant 21210221044.

**Institutional Review Board Statement:** Not applicable.

**Informed Consent Statement:** Not applicable.

**Data Availability Statement:** Not applicable.

**Conflicts of Interest:** The authors declare no conflict of interest.

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
