# Peer review of "Memristive Circuit Design of Nonassociative Learning under Different Emotional Stimuli"

_electronics, doi:10.3390/electronics11233851_

Round 1

Reviewer 1 Report

 Dear authors,

 First of all I would like to congratulate you for this piece of work. It is very interesting and I enccourage you to improve it in orther to be accepted.
- First, I have a general comment regarding the units of some of your equations, as you use a minor 'v' to present the volts, which are usually presented with a 'V' in capital letters. Can you fix this, please?
- Moreover, in order to be able to evaluate the benefits of your proposal, can you present another example of system similar tro yours, please?
- Afterwards, in more detail, can you explain the V1 and V2 values used into your 'stimulus judgment module' (Fig. 4). Can you explain the origin of this values, please?
- On the other hand in Fig. 6 you present the emotion module, but this one is never properly presented in Fig. 3, nor in some other place of your contribution. So, please, can you clarify this?
- Finally, I have two questions abouyt the reliability of the system:
   - Which technology are you planning to use to implement/simulate your system? Device variability is usually a concern for the designers, due to the different behavior of two identical device when are manufactured in electronic devices with highly scaled down dimensions. Then, how do you presume this can affect the performance of your system? Because, by sing small technology nodes (32 or below) variability is gaoining relevance, and it can significantly affect the overall system performance.

A. Asenov et al., “Variability Aware Simulation Based Design- Technology Cooptimization (DTCO) Flow in 14 nm FinFET/SRAM Cooptimization,” IEEE Trans. Electron Devices, vol. 62, no. 6, pp. 1682–1690, 2015, doi: 10.1109/TED.2014.2363117.

But, there are some strategies to manage the expected variability impact in memristive memory systems regarding the different levels of variability in a complex system. So, can this  be of interest of you, can't it?

P. Pouyan, E. Amat, and A. Rubio, “Memristive Crossbar Memory Lifetime Evaluation and Reconfiguration Strategies,” IEEE Trans. Emerg. Top. Comput., vol. 6, no. 2, pp. 207–218, Apr. 2018, doi: 10.1109/TETC.2016.2581700.

- Another reliability question is regarding the endurance of the used devices, because this is a crucial parameter to know the lifetime of your system. Do you have some idea about that?

Author Response

The response to the reviewer's comments has been uploaded to the attachment.

Reviewer 2 Report

The authors present a circuit based on memristors, emulating the nonassociative learning of human brain under emotional stimuli. The work is interesting but the title and the description chosen by the authors is misleading. To my understanding the circuit is simulated with spice using ideal components, without considering the non-idealities of the electonic components. To me this is not a "Memristor-Based Neural Network Circuit" but a "Spice Model for a Memristor-Based Neural Circuit". This is fine but the title of the paper and the corresponding description in the Introduction and in the Conclusions should be modified and clarified. Also remember that the term "neural network" refers to a very specific circuit topology (made by the interconnections of a hyge number of neurons). This is not a neural network but a neural circuit emulating a complex part of our brain. Please modify the paper to clarify it.

Minor issues:

- Please insert at the beginning of section 2 a graphical representation of the physical structure of the memristor. The memristor model uses physical parameters that refer to the structure of the device, but without a picture highlighting them the model is not clearly understandable.

- "The various parameters of the four memristors are shown in Table 1." -> Why four memristors? Please clarify this sentence and correct it.

- What are the voltage levels used in the simulation?

- Please increase the size of the graphs with the results to improve their readability.

Grammatical issues:

- Please rewrite this sentence "The collaboration between neurons can complete the processing and transmission of 14 information, and the concept of artificial neural networks was introduced for information 15 processing.". The second part does not make any sense.

- In this sentence "used in secret communication" what do you mean with "secret", do you mean criptography?

Author Response

(The authors gave the same response as above.)

Round 2

Reviewer 2 Report

The authors have addressed all my comments, the paper is now ready for publication in my opinion.